# Molecular Mechanisms of Renal Progenitor Regulation: How Many Pieces in the Puzzle?

**DOI:** 10.3390/cells10010059

**Published:** 2021-01-02

**Authors:** Anna Julie Peired, Maria Elena Melica, Alice Molli, Cosimo Nardi, Paola Romagnani, Laura Lasagni

**Affiliations:** 1Department of Experimental and Clinical Biomedical Sciences “Mario Serio”, University of Florence, Viale Morgagni 50, 50134 Florence, Italy; mariaelena.melica@unifi.it (M.E.M.); cosimo.nardi@unifi.it (C.N.); paola.romagnani@unifi.it (P.R.); 2Nephrology and Dialysis Unit, Meyer Children’s University Hospital, Viale Pieraccini 24, 50139 Florence, Italy; alice.molli@stud.unifi.it

**Keywords:** renal progenitors, molecular mechanisms, kidney injury, single-cell RNA sequencing, molecular signature

## Abstract

Kidneys of mice, rats and humans possess progenitors that maintain daily homeostasis and take part in endogenous regenerative processes following injury, owing to their capacity to proliferate and differentiate. In the glomerular and tubular compartments of the nephron, consistent studies demonstrated that well-characterized, distinct populations of progenitor cells, localized in the parietal epithelium of Bowman capsule and scattered in the proximal and distal tubules, could generate segment-specific cells in physiological conditions and following tissue injury. However, defective or abnormal regenerative responses of these progenitors can contribute to pathologic conditions. The molecular characteristics of renal progenitors have been extensively studied, revealing that numerous classical and evolutionarily conserved pathways, such as Notch or Wnt/β-catenin, play a major role in cell regulation. Others, such as retinoic acid, renin-angiotensin-aldosterone system, TLR2 (Toll-like receptor 2) and leptin, are also important in this process. In this review, we summarize the plethora of molecular mechanisms directing renal progenitor responses during homeostasis and following kidney injury. Finally, we will explore how single-cell RNA sequencing could bring the characterization of renal progenitors to the next level, while knowing their molecular signature is gaining relevance in the clinic.

## 1. Introduction

Mechanisms of endogenous regeneration and repair have been proposed for several mammalian organs [1]. Classical regenerative organs, such as the gastrointestinal tract and the skin, have been extensively studied over the years and have brought to light the major role of endogenous progenitors [2]. In the intestine, intestinal stem cells maintain daily homeostasis, while distinct stem/progenitor cells are in charge of the fast repair processes following injury [2]. Likewise, epidermal stem cells form a heterogeneous stem cell pool taking part in epidermal homeostasis, as well as tissue repair, following wounding [3].

The adult kidney is an organ with a low cellular turnover and endowed with progenitors capable of proliferating and differentiating [4,5]. This valuable property allows researchers and clinicians to contemplate new therapeutic avenues to restore kidney function after injury.

Here, we propose an overview of the molecular mechanisms taking place in glomerular and tubular renal progenitors in physiological and pathological conditions (Figure 1) and of how a dysregulation of these pathways could be at the origin of kidney disease. We will also examine how renal progenitors could be further characterized using single-cell RNA sequencing (scRNAseq) technology and the clinical relevance of the molecular signature of these cells.

## 2. Renal Progenitors

Renal progenitors were discovered by Sagrinati et al. in human kidneys, based on the expression of the stem cell markers CD133 and CD24, in the absence or low expression of differentiation markers [6,7]. CD133+CD24+ cells are localized at the urinary pole of the Bowman capsule, as well as scattered along the tubular compartment of the nephron among differentiated tubular cells [6]. Some renal progenitors, including those localized in the Bowman capsule and a subset of the ones scattered along the tubule, also express CD106 (also called vascular cell adhesion molecule 1, VCAM1), while the majority of progenitors localized along the tubule do not [6,8]. These phenotypical differences reflect a diverse functional capacity; indeed, CD133+CD24+CD106- cells scattered along the tubules display functional features of tubular progenitors, while CD133+CD24+CD106+ parietal epithelial cells (PECs) are multipotent [6]. In addition, a subset of CD133+CD24+CD106+ progenitors localized close to the distal pole of the Bowman capsule and expressing podocalyxin is able to generate only podocytes [6]. Altogether, these observations configure a hierarchical lineage of renal progenitors within the kidney that reminds the hemopoietic system [9]. PECs with similar progenitor features and anatomical localization were also identified in mouse and rat kidneys [4,10,11]. The genetic tagging of PECs in a transgenic inducible mouse line demonstrated that PECs migrate onto the glomerular tuft and differentiate into podocytes in adolescent mice [11]. More recently, Pax2 has been identified as a marker for mouse renal progenitors, and the creation of an inducible mouse model for lineage tracing of the Pax2+ cell population allowed to demonstrate the differentiation of renal progenitors localized among PECs into podocytes during postnatal glomerular growth [4]. Further studies demonstrated that juxtamedullary and cortical glomeruli have different numbers of Pax2+ progenitors, with cortical ones endowed with twice as many Pax2+ progenitors per glomerular podocyte count in healthy conditions [12]. In adult rat kidneys, immature cells expressing the neural cell adhesion molecule (NCAM) and the progenitor cell marker CD24 have been described among epithelial cells lining the rat Bowman capsule [10].

The genetic tagging of Pax2+ progenitors of the Bowman capsule of mice allowed to demonstrate that these progenitors differentiate into podocytes in models of focal segmental glomerulosclerosis (FSGS), and their response to injury determines the outcome of glomerular disorders, further substantiating their role as podocyte progenitors [4,12]. Recently, using a transgenic mouse model in which podocytes were labeled with GFP (green fluorescent protein) and PECs were simultaneously labeled with tdTomato, Kaverina and colleagues also provided strong evidence that PECs serve as a source of new podocytes in adult mice upon injury. These cells co-expressed the two fluorescent labels, acquired podocyte markers and showed the primary, secondary and tertiary foot processes [13].

An abnormal progenitor response to injury can also contribute to glomerular disorders [4,10,14,15]. Indeed, in certain conditions, in humans, mice and rats, a chaotic migration and proliferation of Bowman capsule progenitor cells has been demonstrated to contribute to crescent formation and glomerular scarring [4,10,14]. Studies on human renal biopsies are consistent with the concept that proliferating progenitors generate hyperplastic lesions in crescentic and collapsing glomerulopathy [14], and similar results have been obtained in rats [10]. In mice, the lineage tracing of PECs demonstrated that their proliferation leads to a marked increase in cell numbers within crescents of murine nephrotoxic serum nephritis and collapsing glomerulopathy [16] and the formation of sclerotic lesions and extracellular matrix deposition in FSGS [15]. More recently, the specific genetic tracking of progenitors among PECs demonstrated their involvement in the generation of hyperplastic glomerular lesions that could be envisioned as a failure to regenerate podocyte following injury [4]. From all these studies, it is now clear that renal progenitors localized among PECs respond to podocyte injury, triggering a regenerative program, but an inefficient or excessive response can lead a functional tissue to become a scar-like tissue composed of cells and disorganized extracellular matrix. Therefore, knowing the mechanisms that drive a correct proliferative and differentiative response of renal progenitors during homeostasis and following injury is of crucial importance and may allow the identification of putative modulators to boost the regenerative potential of renal progenitors.

### 2.1. Regulators of Glomerular Progenitor Physiology: When the Orchestra Tunes the Melody

Which signaling pathways regulate glomerular progenitor quiescence, proliferation and differentiation toward podocytes in healthy kidneys? Studies on nephrogenesis demonstrated that activation of β-catenin/Wnt signaling represents a pivotal step driving PEC differentiation into podocytes during development [17,18]. Indeed, the deletion of Ctnnb1 (β-catenin 1) in PECs in a conditional knockout mouse at the late S-shaped body stage induced glomerular anomalies and the replacement of PECs in Bowman capsules with well-differentiated podocytes. Tracing nephrogenesis in embryonic conditional β-catenin knockout mice revealed that these “parietal podocytes” derived from precursor cells in the parietal layer of the S-shaped body by direct lineage switch. These findings demonstrate that β-catenin/Wnt signaling is required for the proper differentiation and maturation of PECs into podocytes [17]. WT1, a master regulator of this process [19], is also a potent inhibitor of the β-catenin/Wnt signaling pathway [18]. Studies performed in quiescent PECs demonstrated that the expression of WT1 is suppressed by high levels of Pax2 and by the expression of high levels of microRNA-193a (miR-193a) [20]. When PECs downregulate the expression of miR-193a, this allows the upregulation of WT1, which suppress β-catenin/Wnt signaling and induces PEC differentiation into podocytes. Recent in vitro results demonstrated that apolipoprotein L1 (APOL1) also regulates the PEC molecular phenotype through modulation of the miR193a expression and that APOL1 and miR193a share a reciprocal feedback relationship [21]. Indeed, in a culture system, PEC differentiation into podocytes was accompanied by a decrease in miR-193a expression. Similarly, the suppression of miR-193a enhanced the APOL1 expression [21]. Future works should address whether this APOL1–miR-193a axis functions in a similar way in vivo as it does in vitro in relevant transgenic mouse models and in human kidneys. Interestingly, APOL1 is a susceptibility gene, with genetic variants that increase the likelihood to develop podocytopathies [22].

The control of the cell-fate decision and cell proliferation in many different systems is operated through the integrated signaling of the Wnt and the Notch signaling pathways [23]. Lasagni et al. reported that, in renal progenitors localized in the Bowman capsule, Notch activation promotes entry into the S-phase of the cell cycle and subsequent mitosis until they are in an undifferentiated state [24]. However, impaired downregulation of the Notch pathway during renal progenitor differentiation induced the generation of podocytes with abnormal DNA contents and their following deaths by mitotic catastrophe [24,25]. Recent results suggest that podocyte-derived CXCL12 (C-X-C motif chemokine ligand 12) inhibits Notch signaling, thus maintaining the quiescence of podocyte progenitors [12]. Notch downregulation associates with the upregulation of cell cycle inhibitors p21, p27 and p57 and the downregulation of cyclin D1 [24], conferring to the podocyte the characteristics of a postmitotic, nonproliferative cell. The CXCL12-mediated podocyte-renal progenitor feedback mechanism also limits podocyte regeneration after glomerular injury [12]. Indeed, using the lineage tracing of Pax2+ renal progenitors in mice with Adriamycin-induced nephropathy, the researchers showed that a CXCL12 blockade promotes de novo podocyte formation and attenuates glomerulosclerosis [12].

As the enhancement of renal progenitor differentiation into podocytes may represent an attractive therapeutic strategy to promote the remission of glomerular disorders, several studies have been performed to identify differentiating compounds. Retinoic acids (RA) are derivatives of vitamin A with established benefits in the treatment of a variety of cancers [26]. RA have also been shown to protect against renal injury in multiple experimental models of kidney disease, including minimal change disease, membranous nephropathy, FSGS, human immunodeficiency virus (HIV)-associated nephropathy (HIVAN) and lupus nephritis [27]. Numerous studies have underlined the role of RA in podocyte differentiation in vitro [28,29], and we used RA in the cell culture media to promote renal progenitor differentiation towards the podocyte lineage [7]. Interestingly, exposure to albumin, which binds RA with high affinity, during in vitro cultures could inhibit renal progenitor differentiation toward podocytes by sequestering RA. In vivo, we reported that RA were released within the Bowman space following glomerular injury and stopping the endogenous RA synthesis in a model of focal segmental glomerulosclerosis worsened the albuminuria, glomerular injury and mortality [30]. The exogenous administration of RA, neutralizing the sequestering activity of albumin, allowed the regenerative response of renal progenitors, establishing an increase in podocyte number and the improvement of renal function [30]. Recent results from Lasagni et al. [4] corroborated the hypothesis that pharmacological approaches that increase podocyte responsiveness to RA signaling would mitigate the progression of experimental renal injury. Indeed, the in vitro treatment of renal progenitors with RA in the presence of 6-bromo-indirubin-3′-oxime (BIO), a glycogen synthase kinases 3 (GSK3) inhibitor, induced a strong differentiation of human renal progenitors toward podocytes through the activation of RA-responsive elements (RARE) transcriptional activity, i.e., increasing the renal progenitor sensitivity to the differentiating effects of endogenous RA. The enhancement of renal progenitor differentiation into podocytes by using BIO in a murine model of FSGS resulted in an important effect on the disease, increasing the disease remission in treated mice. In a progressive stage mouse model of obesity-related type 2 diabetes, BIO as an add-on to the dual renin-angiotensin system (RAS)/sodium-glucose transporter (SGLT)-2 inhibition with metformin, ramipril and empagliflozin attenuated the glomerular filtration rate (GFR) decline by further reducing glomerulosclerosis, increasing the podocyte numbers through sustaining specialization, as well as inducing de novo differentiation from podocyte progenitors and improving the filtration slit density [31].

Endlich et al. demonstrated the role of Dach1 (Dachshund homolog 1) in the cell fate determination of PEC into podocytes and for proper podocyte function. Podocytes express high levels of Dach1 in vivo and in vitro, while PEC express very low levels of Dach1. The authors found that the induction of Dach1 expression in PEC significantly upregulates the podocyte-specific proteins synaptopodin and WT1. Interestingly, Dach1 is part of the Eya-Six-Hox-Pax regulatory network, and the regulation of synaptopodin expression was accompanied by a concomitant downregulation of Pax2 expression [32].

Guhr et al. analyzed by which mechanisms renal progenitors maintain the potential to express podocyte proteins under pathophysiologic conditions and demonstrated that they contain an activated ubiquitin-proteasome system (UPS) that leads to the rapid degradation of newly synthesized podocyte-specific proteins [33]. On the other hand, the UPS maintains the podocyte identity by regulating the levels of podocyte-specific proteins, such as the actin-binding proteins α-actinin 4 (ACTN4) and synaptopodin (SYNPO), the transcription factor Wilms tumor 1 (WT1), the stomatin family member podocin, the slit diaphragm protein nephrin, the adaptor protein NCK1 and activated protein kinase Cλ (PKCλ) [33]. UPS activity is therefore an important determinant of glomerular cell phenotypes and differentiation status.

It is well-known that, in the kidney, the mechanical environment is subjected to modifications in established models of glomerular diseases and can affect the differentiated state of numerous cell types, including podocytes [34]. We recently analyzed the impact of substrate stiffness on renal progenitor behavior, demonstrating that, at least in vitro, the phenotype of human renal progenitors is highly dependent on the Young’s modulus of the substrate, which is a measure of the stiffness of the material defined as the ratio of stress to strain, with stiffer substrates promoting renal progenitor proliferation and migration. The substrate stiffness modulates also the capacity of renal progenitors to differentiate toward podocytes, with a Young’s modulus of 12 kPa being optimal among those analyzed. Using chemical and genetic inhibitors, we demonstrated that Rho kinase (ROCK) activity is required to mediate the effects of stiffness on renal progenitor proliferation, migration and differentiation [35]. A reduced glomerular stiffness is a common feature of many forms of glomerular injury, including FSGS [34,36], suggesting an important role for ROCK also in kidney disease.

Renin angiotensin aldosterone system inhibitors (RAAS-I) are drugs effective in retarding the progression of kidney disease through a variety of actions. The mechanisms responsible for the therapeutic effects of these drugs, as well as their renal cellular targets, have been largely studied in several animal models of human kidney disease. Recent data demonstrated that they might also exert their beneficial effects by promoting renal progenitor differentiation into podocytes. Indeed, in a rat model of glomerular injury, a treatment with ACE-I induced a reduction of progenitor proliferation, the diminution of crescent formation and avoided the progression toward glomerulosclerosis [10]. Thus, moderation of progenitor cell activation by drugs restored a normal glomerular architecture [10]. Interestingly, the expression of angiotensin (Ang) II receptor, AT1, was limited to rare CD24+ PEC in normal human kidneys but was upregulated in the hyperplastic lesions [37], suggesting a contribution of the Ang II/AT1 receptor pathway in promoting abnormal renal progenitor migration and proliferation in proliferative diseases [37]. In accordance, in a patient affected by CGN (crescentic glomerulonephritis), ACE-I therapy associated with the regression of hyperplastic lesions and normalized the AT1 receptor expression on renal progenitors. These results provide another explanation to the beneficial effects observed after the angiotensin II receptor blocker (ARB) treatment. Similarly, the ARB treatment improved the outcome in a rat model of mesangial proliferative glomerulonephritis, inducing an increase in the number of PECs expressing stem cell markers [38].

Injuries to podocytes are considered an important contributor to diabetic kidney disease progression toward end-stage kidney disease [39,40,41]. Suganami et al. reported the prevention and reversal of renal injury by leptin administration in animal models of diabetic nephropathy [39]. More recently, Pichaiwong et al. demonstrated that replacing leptin could reverse the structural and functional parameters of advanced diabetic nephropathy in leptin-deficient BTBR ob/ob mouse [41]. In particular, the leptin treatment, but not RAAS-I, resulted in a significant increase in podocyte density and number and in an increase of WT1-positive proliferating PEC. The mechanisms underlying this process was further delineated in a follow-up paper, where they showed that a dual treatment of leptin-deficient ob/ob mice with a selective antagonist of the endothelin-1 type A receptor (ETAR) in combination with RAAS inhibition led to an improved phenotype [40], characterized by the activation of PECs and increased number of podocytes. These results provide indirect evidence that PECs may be a potential reservoir to restore lost podocytes and that the differentiative capacity of PECs may be a key element for the regression of diabetic nephropathy that might be pharmacologically modulated.

### 2.2. Regulators of Glomerular Progenitors in Pathology: When the Orchestra Is Out of Tune

While renal progenitors can drive podocyte regeneration following injury [4], they can also originate extracapillary proliferative lesions or crescents that are the hallmark of both inflammatory and noninflammatory glomerular diseases [42]. Indeed, evidence in experimental models [15] and in human biopsies indicate that crescents are composed of renal progenitors [14] that abnormally shift their reactions from reparative to injurious. It is not completely understood which factors are responsible for tilting the balance. CGN is the best-characterized disease in which renal progenitors are the major culprits. Cellular crescent is the typical morphological change observed in CGN. It is defined as the multilayered accumulations of renal progenitors and other cell types within the Bowman space. Consequently, it occludes the urinary outlet and the flow of the primary urine, and later, the implicated nephron is impaired. The rupture of glomerular capillaries in crescentic disease leads to the exposure of renal progenitors to a high concentration of plasma that dramatically increases the proliferation of human renal progenitors in culture [43]. Several plasma components can account for the crescent formation, but, currently, there are consistent data only for fibrinogen activation, a member of the activated coagulation cascade during vascular injuries. A lack of fibrinogen or fibrinolysis prevents crescent formation in several rodent models [43,44].

Collapsing nephropathy and pseudocrescents also originate from renal progenitors [14]. At difference with crescents, it was proposed that pseudocrescents originate from renal progenitors as a dysregulated response to the massive and fast podocyte detachment occurring in certain conditions of direct podocyte injury (such as exposure to certain drugs, immune-mediated disorders or infections that directly target the podocyte) occurring in the absence of inflammatory components and leading to capillary collapse [22,45]. These lesions are also frequently observed in viral glomerulopathies, such as HIV- and parvovirus-nephropathy [22]. In these viral glomerulopathies, interferon (IFN-)-α and IFN-β not only trigger local inflammation inside the glomerulus but, also, act on PECs and podocytes, with IFN-α inhibiting the migration of PECs and both suppressing renal progenitor differentiation into podocytes in vitro [46]. In vivo, in a model of Adriamycin nephropathy, the injection of either IFN-α or IFN-β aggravated proteinuria and glomerulosclerosis [46]. Recently, collapsing FSGS has been described in patients of recent African ancestry with high-risk APOL1 genotype and infected with severe acute respiratory syndrome coronavirus 2 (SARS-CoV-2) [47,48]. It has been proposed that SARS-CoV-2 could directly infect the podocyte [49] and/or trigger an inflammatory cascade that involves activation of the interferon–chemokine pathway, which, in turn, interacts with the APOL1 variant gene [50]. As indicated above, renal progenitor differentiation into podocytes associates with APOL1 expression and could therefore be involved in coronavirus disease 2019 (COVID-19)-associated nephropathy.

Several recent studies highlighted a critical role for the de novo expression of CD9 and, subsequently, of CD44 as a pathogenic switch of PECs from a quiescent to an activated phenotype in CGN and in FSGS [16,51,52], confirming the pathogenic role of PECs in these diseases and offering new molecular targets for glomerular disease therapy. In support of this idea, Kaverina et al. showed that PECs lose CD44 expression when differentiating into podocytes in injured glomeruli of old mice, suggesting that a CD44 increase in PECs represents not a regenerative but a pathological transition [53]. In FSGS, CD44 has been shown to have an important role in cell migration toward the injured filtration barrier, where injured podocytes upregulate the migration inhibitory factor (MIF) and stromal cell-derived factor 1 (SDF1) that stimulate CD44 expression and CD44-mediated migration [54]. Additionally, PECs produced both PEC-derived and podocyte-specific extracellular matrix protein isoforms in a CD44-dependant manner [55]. Finally, a lineage tracing study of PECs suggested that CD44 did not take part in kidney regeneration through differentiation into podocytes and only participated in a pro-fibrotic pathway [56].

## 3. Tubular Progenitors

Renal progenitors from the parietal epithelial layer of the Bowman capsule can potentially regenerate proximal tubular epithelial cells at the glomerulotubular junction [57]. However, tubular-committed progenitors scattered in the proximal and distal tubules also exist in humans [6,58,59,60] and in mice [5,61,62,63] and increase upon tubular injury in patients affected with acute or chronic tubular damage [6].

Kumar et al. performed lineage tracing of rare Sox9-expressing cells in the proximal tubule and identified them as a putative tubular progenitor population involved in post-acute kidney injury (AKI) recovery [64]. Sox9 is a transcription factor that, in kidney development, controls epithelial branching and is expressed within nephron precursors [64,65]. Interestingly, when Sox9 was knocked out from the S1 and S2 segments, a slower recovery of the physiological renal functions, enhanced tubular injury, as well as increased renal fibrosis, occurred [64]. After partial nephrectomy, Sox9+ cells proliferate and generate epithelial cells of the proximal tubule, Henle’s loop, distal tubule, collecting duct and the parietal layer of glomerulus [66].

Recently, Lazzeri et al. provided evidence that tubular progenitors undergo mitosis and replace approximately half of the irreversibly lost tubular cells during AKI [5]. Performing lineage tracing of Pax2+ cells in a mouse model of tubular injury, they identified tubular progenitors as a distinct tubular cell subpopulation that was resistant to death and displayed high clonogenic activity, leading to the generation of long tubule segments [5].

### 3.1. Regulators of Tubular Progenitor Physiology: A Polyphonic Choir

Human renal progenitors express B lymphoma Mo-MLV (Moloney murine leukemia virus) insertion region 1 (Bmi-1) [57]. Bmi-1 is a member of the polycomb family of transcriptional repressors. It is involved in cell cycle regulation and the senescence of stem cells endogenous to various organs, such as the prostate, small intestine and lungs [67,68,69,70]. In the kidneys, BMI-1 levels increased rapidly following injury in a mouse model of AKI [71]. These findings point toward the involvement of Bmi-1 expressed in tubular progenitors in renal regeneration. Indeed, Lv et al. showed that acute tubular necrosis led to a Bmi-1 increase and subsequent tubular progenitor mobilization in wild-type mice, while tubular progenitors were not mobilized in Bmi-1 knockout mice [72]. Bmi-1 knockout mice displayed a strong renal phenotype, including interstitial fibrosis, tubular atrophy and severe renal dysfunction, with decreased cell proliferation, increased cell apoptosis and senescence and inflammatory cell infiltration [72,73]. In a recent study, Zhou et al. further elucidated the role of Bmi-1 in renal progenitors, showing that Bmi-1 preserved the self-renewal and stemness of renal progenitors by maintaining the redox balance and preventing cell cycle arrest, through the inhibition of reactive oxygen species (ROS), p16 and p53 [74].

Another important molecule involved in the regulation of the tubular progenitor is Toll-like receptor 2 (TLR2), or CD282, an evolutionary conserved membrane protein that plays an important role in pathogen recognition and the activation of innate immunity. TLR2 acts as a sensor of tissue injury through the detection of damage-associated molecular pattern molecules (DAMPs) released by damaged tissues. TLR2 activation leads to the activation of downstream transcription factors that regulate the expression of survival genes or proinflammatory cytokines and chemokines [75,76,77]. Sallustio et al. showed that tubular progenitors express TLR2, whose stimulation by agonists that mimic inflammatory mediators or DAMPs induces the massive secretion of monocyte chemoattractant protein-1 (MCP-1), interleukin 6 (IL-6), interleukin 8 (IL-8) and complement component C3 via NF-κB (nuclear factor kappa-light-chain-enhancer of activated B cells) activation [59]. Moreover, TLR2 stimulation modulated the proliferation rate and differentiation capacity of tubular progenitors, suggesting an important role in renal repair [59]. Follow-up studies by the same group identified distinct sets of miRNAs specifically expressed in tubular progenitors [78]. Among those, miR-1915 and miR-1225-5p regulated the expression of CD133 and PAX2, as well as TLR2. Sallustio et al. then dissected the recovery mechanisms following AKI and found an essential role for TLR2 in renal regeneration [79]. They established that, following injury, TLR2 damage sensing leads to the secretion of inhibin-A and decorin by the tubular progenitors, which, in turn, promote tubular regeneration through cell proliferation [79]. These two cytokines belong to the TGF-β (transforming growth factor-β) signaling pathway and are involved in cell cycle regulation, the increase of cell proliferation and the inhibition of apoptosis [80,81,82,83,84].

The expression of molecules from the Wnt pathway has been reported in adult renal progenitors in mice [85] and humans [86]. Using a mouse model of lineage tracing, Rinkevich et al. showed that, both during homeostasis and following injury, adult mammalian kidneys undergo segment-specific clonal expansion from cells derived from WNT responsive precursors [63]. They suggested that the ability to respond to WNT signals selects for the cells which will ultimately carry out robust clonal expansion. Studies in SIX2+ urine-derived renal progenitors indicated that WNT pathway activation by GSK3β inhibition induces the differentiation of renal progenitors into renal epithelial proximal tubular cells [87]. In addition, Wnt3 exerted pro-regenerative effects and was upregulated in CD133+ renal progenitors in an in vitro model of cisplatin injury [88]. In this study, the authors unveiled the functional role of CD133 itself in renal tubular repair through the maintenance of the proliferative response and control of senescence by acting as a permissive factor for β-catenin signaling, preventing its degradation in the cytoplasm [88]. In zebrafish kidneys, damaged tubules were replaced by new nephrons from renal progenitors expressing the Wnt receptor frizzled9b and the transcription factor lef1. Following injury, the expression of Wnt ligands Wnt9a and Wnt9b was induced in injured kidneys at sites where the progenitor cells form new nephrons [89]. These results suggest that the essential role of the Wnt/frizzled signaling pathway in kidney regeneration is highly conserved among species.

As previously mentioned, Notch signaling is an evolutionary conserved pathway that has a critical role in kidney injury and repair [24,90,91,92,93], particularly during AKI [94,95]. Kang et al. showed that Sox9+ renal progenitors expressed high levels of Notch, and overexpression of the Notch1 intracellular domain (NICD1) in the Sox9+ population improved the renal histology in a folic acid-induced model of AKI [62]. Ma et al. reported that the activation of Sox9+ renal progenitors, whose role is essential in kidney repair, was mediated by the Notch pathway, confirming previous report that the Notch1-3, Jagged1/2, Dll4 and Sox9 expression levels increase after ischemia-reperfusion injury (IRI) [66]. Indeed, in other organs such as the pancreas, Sox9 activation modulates the Notch pathway by regulating Hes1 to maintain the progenitor cell pool [96].

Several drugs have been shown to improve kidney regeneration, and, among those, histone deacetylase (HDAC) inhibitors (HDACis) may be a promising therapeutic option for the treatment of AKI [97,98,99,100,101,102]. HDACs form a group of enzymes involved in multiple cellular processes by removing the acetyl group from histone or nonhistone proteins [103]. Marumo et al. reported a reduction in HDAC5 activity, increased histone acetylation and reactivation of bone morphogenetic protein 7 (BMP-7) in proximal tubular cells during the recovery phase following renal IRI [104]. These observations suggest that HDACis might exert their beneficial effects on renal recovery through the increased expression of BMP-7, a protein that maintains a renal progenitor pool in undifferentiated status during kidney development [105]. Interestingly, the treatment with HDACis expanded the renal progenitor cell population in zebrafish [106]. In the nephrotoxic serum nephritis model of glomerulonephritis in mice, a trichostatin A (TSA) treatment activated kidney side population (SD) cells [107]. SD cells form a subset of cells with multilineage potential and known renoprotective properties that attenuate chronic kidney disease (CKD) through an increase of BMP-7 expression [107]. Using the lineage tracing approach described above, Lazzeri et al. showed that a treatment with two widely used HDACis, TSA and 4-phenylbutyrate (4-PBA), led to Pax2+ progenitor proliferation, consequently avoiding the development of tissue fibrosis and CKD [5]. The development of selective HDACis, with enhanced efficacy and less toxicity, would improve kidney recovery through tubular progenitor proliferation. Of note, HDACis have shown beneficial therapeutic effects in numerous experimental models of kidney diseases besides AKI, including glomerulosclerosis, tubulointerstitial fibrosis, glomerular and tubulointerstitial inflammation, lupus nephritis, polycystic kidney disease and renal cell carcinoma (RCC), as reviewed in [108]. Several HDACis are currently in Phase 1 or 2 trials for the treatment of RCC and renal impairment (clinicaltrial.org).

### 3.2. Regulators of Tubular Progenitors in Pathology: A Cacophonus Choir

Biological and molecular features of kidney cancer suggest that renal progenitors could be at the origin of the development of different kidney tumor types.

In a recent study, Peired et al. showed that human renal progenitors overexpressing NICD1 had an increased proliferative capacity and form aberrant mitosis in 2D cultures and could generate a tumor-like mass in 3D cultures [8]. Similarly, Pax2+ renal progenitors overexpressing NICD1 following transgene induction in adult mice or following IRI were at the origin of papillary adenomas and RCCs [8]. In confirmation of this finding, a treatment blocking endogenous AKI-induced NOTCH1 activation led to the development of fewer tumors [8].

Recently, two studies suggested that angiomyolipomas originate from multipotent kidney epithelial cells localized in the tubule and undergoing clonal expansion in response to tuberous sclerosis complex (TSC) gene deletion [109,110]. Both studies proposed these cells could be renal progenitors with multilineage differentiation capacity [109,110]. Interestingly, Cho et al. revealed that the activation of a previously unreported Rheb-Notch-Rheb regulatory loop, in which the cyclic binding of Notch1 to the Notch-responsive elements (NREs) on the Rheb promoter is a key event, was the main mechanism behind the generation of the multiple lineages present in angiomyolipoma [109]. Taken together, these results indicate that a deregulation of the Notch pathway in renal progenitors can lead to renal pathologies.

Wan et al. observed that SOX9 expression was upregulated in RCC patients and correlated with the advanced pathological grade [111]. RCC patients with high SOX9 levels also had shorter survival [111]. These data confirmed a precedent study that associated SOX9 expression with RCC Fuhrman grading and showed that patients with SOX9 (−) had a much better therapeutic response to tyrosine kinase inhibitors than those with SOX9 (+) [112]. Therefore, we could hypothesize that an increase of SOX9 expression in SOX9+ renal progenitors could contribute to RCC development. A similar mechanism was described in basal-like breast cancer, where SOX9 expression in luminal stem/progenitor cells could control the lineage plasticity for cancer through the activation of NF-κB signaling [113].

## 4. Outlook on the Future of Renal Progenitors

### 4.1. Single-Cell RNA Sequencing: Let Us Get in Tune with the Times

The fast development of scRNAseq is opening new perspectives for dissecting the molecular processes involved in renal progenitor regulation in physiological and pathological conditions. ScRNAseq consists in obtaining gene expression profiling at a single-cell resolution, putting in evidence the different cellular states and molecular dynamics of even the rarer subpopulations. This novel technology has been used successfully in several organs—for example, to study Prominin 1+ liver progenitors [114], Dach1–downregulated lymphoid progenitors [115] and KTR5+ lung progenitors in COVID-19 patients [116]. Within the past few years, an increasing number of research groups have applied this strategy to define the cell populations of the kidneys in mice and humans [117,118,119,120]. In a very recent study, Rudman-Melnick et al. identified the transcriptional signature of all cell populations in an experimental model of AKI, highlighting the presence of previously undescribed injury-related molecules [119]. Such an approach could potentially reveal novel mechanisms activated in renal progenitors following AKI, leading to the identification of potential molecular targets. In their seminal paper, Young et al. were able to match clear cell and papillary RCC cells to a subtype of proximal convoluted tubular cells defined by SLC17A3 and VCAM1 expression [117]. As mentioned earlier, VCAM1 or CD106 expression characterizes, together with CD133, a rare population of renal progenitors scattered mostly in the proximal tubule [6]. An analysis of the scRNAseq data revealed that the human renal progenitor transcriptome shows similarities to PT1, the putative cell of origin of human papillary RCC [8]. These observations substantiate our hypothesis that papillary RCC originates from the Notch-mediated transformation and proliferation of a proximal tubule population of renal progenitors [8].

### 4.2. Clinical Applications: The Clinic Calls the Tune

Renal progenitor-based therapies represent a promising new frontier in the treatment of renal diseases, as several studies suggest that they improve kidney function following injury [121]. However, injecting renal progenitors directly into animal models of kidney injury to induce tissue regeneration presents important limitations that have been exposed elsewhere [121]. These caveats could be circumvented thanks to the newly exploited properties of renal progenitors, which is their capacity to secrete trophic factors, cytokine or chemokines that efficiently mediate kidney repair in a paracrine or autocrine manner (Figure 2). Kenji et al. reported that the intraperitoneal injection of culture supernatant obtained from adult rat kidney progenitors significantly suppressed the tubular cell apoptosis of residual renal cells, diminished the inflammation and promoted the proliferation of immature cells in an experimental IRI model through the release of HGF (hepatocyte growth factor), EGF (epidermal growth factor), TGF-β and Epo (erythropoietin) [122]. Indeed, the therapeutic use of numerous growth factors has been reported to ameliorate kidney injuries, such as HGF, BMP7, EGF, TGF-β and VEGF (vascular endothelial growth factor) [123,124,125,126,127]. Sallustio et al. reported that human renal progenitors not only significantly repair damage tubular cells but, also, exhibit antifibrotic effects via the secretion of CXCL6 (C-X-C motif chemokine ligand 6), SAA2 (serum amyloid A2), SAA4 (serum amyloid A4) and BPIFA2 (BPI (bactericidal permeability-increasing) fold-containing family A member 2) through a paracrine mechanism [128]. Aggarwal et al. reported that the renal progenitor secretion of Epo limits renal fibrosis after tubular injury [129]. In addition to soluble factors, renal progenitors secrete extracellular vesicles (EVs), nanometer-sized lipid bilayer-delimited particles carrying bioactive lipids, proteins and RNAs that allow cell-to-cell communication through paracrine actions. The smallest and best-described type of EVs are the exosomes, which have been recently investigated for their protective effects against IRI-induced AKI [130,131]. Li et al. demonstrated that renal progenitor-derived exosomes could restore renal structures and functions via their immunomodulatory, antiapoptotic and proliferation stimulation abilities in AKI models. MicroRNAs (miRNAs) were the most abundant components of the exosomes, and among those, miR-146a was identified as the key player in mediating cytoprotective effects by downregulating IRAK1 (interleukin 1 receptor associated kinase 1)/NF-kB signaling [130]. In a model of diabetic nephropathy, urinary progenitor-secreted exosomes were found to reduce podocyte apoptosis by suppressing caspase-3 and promoting vascular regeneration, which may be related with the cytokines VEGF, TGF-β1, angiogenin and BMP-7 contained in urinary progenitor-derived exosomes [132]. The inhibition of podocyte apoptosis was also related to the overexpression of miR-16-5p in urinary progenitor-secreted exosomes by suppressing VEGF-A [133].

Molecules and exosomes secreted by renal progenitors promote recovery from kidney disease through their ability to exert a series of renoprotective and regenerative effects thanks to their reduced immunogenicity and lower risk of maldifferentiation and tumorigenesis compared to cell therapies. These remarkable features make them appealing for clinical applications.

Another challenge in the clinical approach to kidney disease is the discovery of new tools for diagnosing and monitoring kidney disease that would be easily accessible with noninvasive procedures. In this context, urine represents a valuable biofluid due to its accessibility, fast and easy sampling and broad variety in proteins, metabolites, cells and cellular contents released from the urogenital tract [134]. The presence of cells in urine that display stem cell properties was first described by Zhang et al. in 2008 [135]. In the following years, several groups developed techniques to isolate and characterize urine-derived progenitors from healthy donors and patients with kidney disorders [136]. No formal consensus has yet been reached on which markers may be used to define urine-derived progenitors. Most studies indicated that they express mesenchymal stem cell markers (CD44, CD73 and VIM) and stem cell markers (such as POU5F1, SSEA4 and TRA-1-81, as well as CD117) but no markers of hematopoietic- or urothelium-derived cell lineages and low levels of tubular- or podocyte-specific markers [136]. Regarding their origins, Bharadwaj et al. showed that urine-derived renal progenitors carried the Y chromosome in a male-to-female kidney transplant recipient, indicating that they come from the kidneys [137]. These cells have the capability to differentiate into podocytes [138] and express podocyte- and PEC-specific protein markers [137,139], suggesting that they originate from PECs. A comparative transcriptome analysis of urine-derived renal progenitors and kidney biopsy-derived renal epithelial proximal cells confirmed the renal progenitor identity of urine-derived progenitors [87], indicating that they could also originate from scattered tubular progenitors. These cells can be reprogrammed into induced pluripotent stem cells (iPSC) and be used for regenerative medicine, disease modeling or pharmacological testing [140,141].

Recently, it has been proposed that the expression of the renal progenitor marker CD133 in urinary EVs represents a good marker for the evaluation of the functional status of the renal tubular compartment and of the presence of cells with proliferative and repairing activity within tubules after injury. Indeed, two studies reported that CD133+ urinary EV levels, elevated in healthy subjects, not only decrease in patients with acute tubular damage [142] but, also, in acute and chronic glomerular conditions [143]. Furthermore, the presence of renal progenitors themselves in urine may reflect the pathophysiological status of renal tissue. In particular, Manonelles et al. provided evidence that the isolation of CD133+ CD24+ renal progenitors from the urine of stable allograft recipients at six months could predict the poor long-term outcome of the transplant at two years [144]. Renal progenitor proliferation and migration from the Bowman capsule to the glomerular tuft across the urinary space in order to replace detached podocytes could explain the excretion of renal progenitors and, if sustained over time, might fail to preserve the allograft function, resulting in GFR decline, albuminuria and chronic glomerular histological lesion development [144].

Urine-derived kidney cells could also be a powerful personalized tool for functional studies on candidate variants in inherited renal disease. As described by Lazzeri et al., urine-derived renal progenitors obtained from patients carrying pathogenic mutations in genes encoding for podocyte proteins expand in culture but develop anomalies in the expression or localization of podocyte proteins following podocyte differentiation [138]. In agreement with this evidence, the same technique was used to demonstrate the pathogenicity of a NPHS1 gene variant of unknown significance in a patient with refractory lupus nephritis [145]. Another study underlined the possibility to use urine-derived renal epithelial cells to carry out RNA and functional studies on kidney-specific genes, validating the pathogenicity of a synonymous variant in PKHD1 (polycystic kidney and hepatic disease 1) and confirming the genetic diagnosis of ARPKD (Autosomal Recessive Polycystic Kidney Disease) in a patient with CKD associated with atypical polycystic kidneys [146].

## 5. Conclusions

A vast body of literature describes the numerous mechanisms of the regulation of renal progenitors in the glomerular and in the tubular compartments, allowing us to have a global vision of the complexity of the molecular processes taking place in the physiological and in pathological conditions. Knowing the molecular signature of renal progenitors opens the door to identifying new targets for drugs to sustain kidney regeneration or biomarkers to monitor kidney health.

## Figures and Tables

**Figure 1 cells-10-00059-f001:**
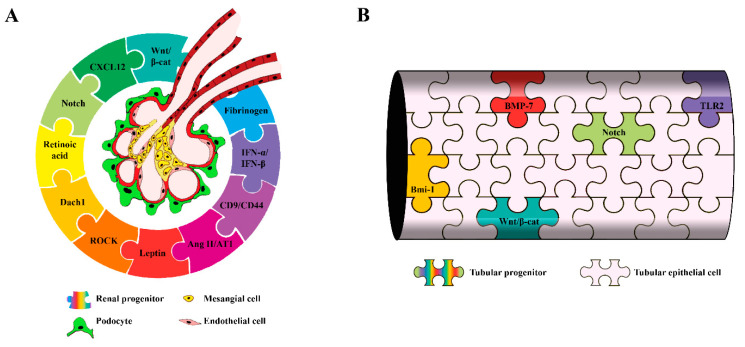
Main molecular mechanisms controlling renal progenitor responses in physiological and pathological conditions: (**A**) glomerular progenitors and (**B**) tubular progenitors. β-cat: β-catenin, CXCL12: C-X-C Motif Chemokine Ligand 12, Dach1: Dachshund homolog 1, ROCK: Rho kinase, Ang II/AT1: Angiotensin II/Angiotensin II receptor, IFN-α/IFN-β: interferon-α and interferon-β, BMP-7: bone morphogenetic protein 7, Bmi-1: B lymphoma Mo-MLV insertion region 1 and TLR2: Toll-like receptor 2.

**Figure 2 cells-10-00059-f002:**
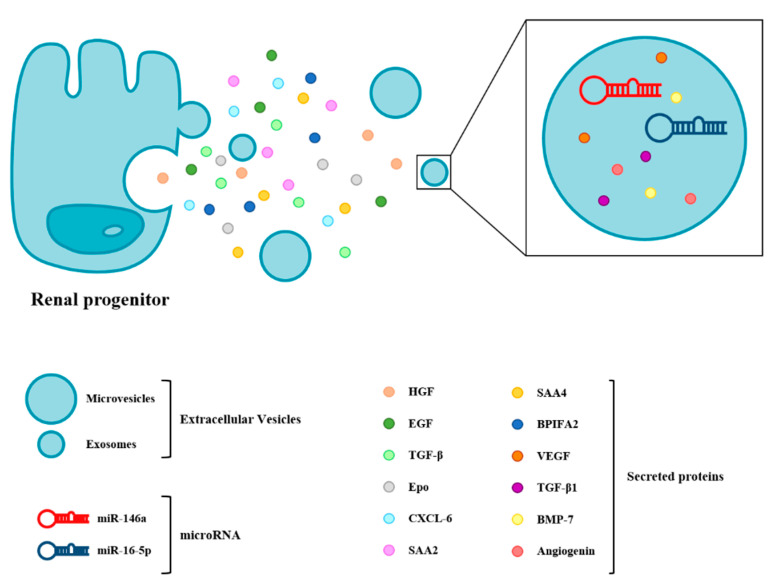
Adult kidney progenitor cells secrete soluble molecules, as well as molecule-containing extracellular vesicles, that contribute to the regeneration of the kidneys. miR: microRNA, HGF: hepatocyte growth factor, EGF: epidermal growth factor, TGF-β: transforming growth factor-β, Epo: erythropoietin, CXCL6: C-X-C motif chemokine ligand 6, SAA2: serum amyloid A2, SAA4: serum amyloid A4, BPIFA2: BPI (bactericidal permeability-increasing) fold-containing family A member 2, VEGF: vascular endothelial growth factor and BMP-7: bone morphogenetic protein 7.

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
