# Peer review of "Molecular Mechanisms of Renal Progenitor Regulation: How Many Pieces in the Puzzle?"

_cells, 2021, doi:10.3390/cells10010059_

Round 1

Reviewer 1 Report

This is a useful and timely review. The suggestions I have are quite small and would be easy to address.

1) In the section on crescents and collapsing glomerulopathy's (lines 90 through 108) it would be valuable to developin more detail the distinction between the responses that lead to podocyte hypertrophy and hyperplasia in collapsing glomerulopathy versus the formation of crescents. The current section is somewhat sketchy and does not clearly establish differences for the roles of progenitor cells in these distinct entities

2) Line 202. What is Young's modulus?

3) Lines 253-8. It would be current and useful to mention Covid -related collapsing glomerulopathy when discussing other infectious ideologies of collapsing glomerulopathies.

4) Lines 259-265. Can the authors provide some judgment as to whether the expression of CD44 on parietal epithelial cells is indicative of an activation state that leads to the transformation of progenitor cells to newly generated podocytes or potentially other cell types in the glomerulus, versus serving as an injury and migratory marker without a specific role in cellular regenerative processes.

Author Response

This is a useful and timely review. The suggestions I have are quite small and would be easy to address.

We thank the reviewer for his/her kind comments. We checked our manuscript for English language and style.

1) In the section on crescents and collapsing glomerulopathy's (lines 90 through 108) it would be valuable to develop in more detail the distinction between the responses that lead to podocyte hypertrophy and hyperplasia in collapsing glomerulopathy versus the formation of crescents. The current section is somewhat sketchy and does not clearly establish differences for the roles of progenitor cells in these distinct entities

We thank the reviewer for this observation. We rephrased this section and added another one page 6 lines 245-250.

2) Line 202. What is Young's modulus?

We apologize for the lack of clarity. We included the definition of Young’s modulus page 5 lines 191-192.

3) Lines 253-8. It would be current and useful to mention Covid -related collapsing glomerulopathy when discussing other infectious ideologies of collapsing glomerulopathies.

Indeed, several cases of collapsing glomerulopathies have been reported in COVID-19 patients, we apologize for the omission. We added a new paragraph page 6 lines 255-261.

4) Lines 259-265. Can the authors provide some judgment as to whether the expression of CD44 on parietal epithelial cells is indicative of an activation state that leads to the transformation of progenitor cells to newly generated podocytes or potentially other cell types in the glomerulus, versus serving as an injury and migratory marker without a specific role in cellular regenerative processes.

We thank the reviewer for this constructive request, we addressed it page 6 lines 268-274.

Reviewer 2 Report

The review by Peired and colleagues is very comprehensive, up to date, and covers aspects of the kidney regeneration field in detail without bias. References are well cited, and are inclusive. I have no comments that would improve this fine paper.

Author Response

The review by Peired and colleagues is very comprehensive, up to date, and covers aspects of the kidney regeneration field in detail without bias. References are well cited, and are inclusive. I have no comments that would improve this fine paper.

We thank the reviewer for his/her kind comments.

Reviewer 3 Report

The authors have carefully and systematically presented a comprehensive review on currently known molecular mechanisms of renal progenitor regulation.

This comes as no surprise as the authors are among the top groups working on kidney stem cells.

It was indeed a pleasure and educational reading this review

My only request is that the authors should dedicate a short section describing what is current on the isolation of renal progenitor cells from human urine. They indeed cite a publication as ref.79. My guess is that there are indeed additional similar reports.

Some claim that urine-derived renal progenitor cells can only be isolated from individuals with some sort of renal injury or disease, true or false?

They should also to the best of their knowledge shed light on the origins (where within the kidney) of these urine-derived progenitor cells. Do they originate from PECs?

Finally, the authors should lay out standards for the characterisation of these cells urine-derived renal progenitor cells. 

Author Response

The authors have carefully and systematically presented a comprehensive review on currently known molecular mechanisms of renal progenitor regulation.

This comes as no surprise as the authors are among the top groups working on kidney stem cells.

It was indeed a pleasure and educational reading this review

We thank the reviewer for his/her kind comments. We checked our manuscript for English language and style.

My only request is that the authors should dedicate a short section describing what is current on the isolation of renal progenitor cells from human urine. They indeed cite a publication as ref.79. My guess is that there are indeed additional similar reports.

We thank the reviewer for his interesting request. We added a new section pages 11-12 lines 468-511, dedicated to urine-derived progenitor cells and their utility in the clinic. In particular, we underlined the importance of these cells to evaluate kidney function and to study genetic disorders.

Some claim that urine-derived renal progenitor cells can only be isolated from individuals with some sort of renal injury or disease, true or false?

A vast body of literature documents the possibility to isolate urine-derived progenitors from both healthy donors and patients with kidney disorders (see page 11 line 473-474). However, depending on the urine source and isolation protocol, it might be difficult to obtain cultures from healthy individuals (Lazzeri et al, reference [5]).

They should also to the best of their knowledge shed light on the origins (where within the kidney) of these urine-derived progenitor cells. Do they originate from PECs?

We thank the reviewer for pointing out this intriguing question. We addressed it page 11 lines 479-486.

Finally, the authors should lay out standards for the characterisation of these cells urine-derived renal progenitor cells. 

When thank the reviewer for this suggestion and we added the standard for the characterization of urine-derived progenitors page 11 lines 475-479.